# Altering the Chain Length Specificity of a Lipase from *Pleurotus citrinopileatus* for the Application in Cheese Making

**DOI:** 10.3390/foods11172608

**Published:** 2022-08-28

**Authors:** Niklas Broel, Miriam A. Sowa, Julia Manhard, Alexander Siegl, Edgar Weichhard, Holger Zorn, Binglin Li, Martin Gand

**Affiliations:** 1Institute of Food Chemistry and Food Biotechnology, Justus Liebig University Giessen, 35392 Giessen, Germany; 2Optiferm GmbH, 87466 Oy-Mittelberg, Germany; 3College of Food Science and Engineering, Northwest University, Xi’an 710069, China

**Keywords:** *Pleurotus citrinopileatus*, lipase, semi-rational design, cheese making, chain length specificity

## Abstract

In traditional cheese making, pregastric lipolytic enzymes of animal origin are used for the acceleration of ripening and the formation of spicy flavor compounds. Especially for cheese specialities, such as Pecorino, Provolone, or Feta, pregastric esterases (PGE) play an important role. A lipase from *Pleurotus citrinopileatus* could serve as a substitute for these animal-derived enzymes, thus offering vegetarian, kosher, and halal alternatives. However, the hydrolytic activity of this enzyme towards long-chain fatty acids is slightly too high, which may lead to off-flavors during long-term ripening. Therefore, an optimization via protein engineering (PE) was performed by changing the specificity towards medium-chain fatty acids. With a semi-rational design, possible mutants at eight different positions were created and analyzed in silico. Heterologous expression was performed for 24 predicted mutants, of which 18 caused a change in the hydrolysis profile. Three mutants (F91L, L302G, and L305A) were used in application tests to produce Feta-type brine cheese. The sensory analyses showed promising results for cheeses prepared with the L305A mutant, and SPME-GC-MS analysis of volatile free fatty acids supported these findings. Therefore, altering the chain length specificity via PE becomes a powerful tool for the replacement of PGEs in cheese making.

## 1. Introduction

The formation of flavor compounds in cheese is a highly complex and versatile process. Different catabolic reactions, mainly lipolysis and proteolysis, lead to the formation of piquant cheesy flavors [1]. For certain types of cheese, especially for white cheeses, so-called brine cheeses, such as Feta and its variants created from cow’s milk, for kashkaval and many other Italian cheeses, such as *Parmigiano Reggiano* or Provolone, the majority of character impact compounds are short- to medium-chain free fatty acids such as butanoic acid (C4:0), hexanoic acid (C6:0), octanoic acid (C8:0), or decanoic acid (C10:0) [2]. Milk fat contains more than 400 different fatty acids, of which only 15 are present in amounts above 1% of the total fatty acids, chemically bound in triglycerides [3]. However, only volatile free fatty acids (vFFA) contribute to the flavor. These vFFA possess different olfactory properties depending on their chain lengths, ranging from goaty, cheesy (e.g., C6:0, C8:0, C10:0) to soapy, waxy (e.g., C12:0, C14:0) flavors [4]. The formation of these vFFA is catalyzed by lipolytic enzymes during the ripening process. Endogenous lipases in milk were originally used to produce this special type of flavor, but due to the pasteurization of the raw milk, which is the industrial standard to inactivate pathogens and other microorganisms to increase the shelf life of the products, those enzymes are inactivated. To compensate for this heat damage, enzyme preparations may be added. Most commonly used enzymes are derived from goats, sheep, or calves in the form of extracts of sublingual tissue, so-called pregastric esterases (PGE), or paste rennet from the stomachs of the aforementioned animals [5,6,7]. PGEs show a high specificity towards short-chain fatty acids (C4:0 to C10:0), which enables them to create goaty, cheesy flavors, without adding a soapy note caused by the hydrolytic release of long-chain fatty acids [8]. However, these animal-derived enzyme preparations are neither suitable for a vegetarian, kosher, or halal diet, nor for the complete value chain of the side stream product whey. Therefore, alternative enzyme preparations are highly sought-after. To overcome these limitations, the market offers numerous lipolytic enzymes from different origins such as a lipase from *Rhizomucor miehei,* which is sold under the tradename Piccantase^®^. Nevertheless, these enzymes do not meet all requirements by either lacking the functionality in terms of producing a flavor that is typical for cheese prepared with PGE or tending to release lipids non-specifically, which may lead to soapy off-flavors. Therefore, finding suitable replacements for PGEs is challenging [8,9]. Despite these challenges, a suitable candidate was found in a previous study, as described by Sowa et al. [10]. Application tests with different culture supernatants of basidiomycetous fungi revealed that these kinds of enzyme preparations may represent suitable replacements for PGEs. With the purified lipase of the golden oyster mushroom, *Pleurotus citrinopileatus* (PCI_Lip), the sensory properties of the produced cheese were close to those of a cheese prepared with a PGE reference enzyme (optizym z10uc). Unfortunately, these cheeses still had a slight soapy aftertaste [10]. Therefore, a semi-rational design was used as a powerful tool in the present study to engineer the PCI_Lip chain length specificity, in particular, to reduce the hydrolysis of milk triglycerides with long-chain fatty acids. The semi-rational design combines the advantages of both, directed evolution and rational design, by the usage of partial knowledge of the protein structure without the need for a crystal structure. Smart libraries can be created and analyzed in silico, providing smaller and easier-to-screen libraries compared to directed evolution approaches [11].

Most lipases belong to the group of α/β-hydrolase fold proteins. α/β-Hydrolases can be further classified according to a few structural motifs in addition to their core catalytic domain, the existence of a lid, or a cap, N- and C-terminal domains and various combinations thereof. The lid is a hydrophobic domain located above the active site and the substrate channel of the enzyme and is responsible for the interfacial activation of certain lipases [12]. Today, various classification systems for lipases exist. A structure-based classification according to the Lipase Engineering Database (LED v4.1, https://led.biocatnet.de, accessed on 1 May 2022) represents the most commonly used tool. The database consists of 280,638 protein sequences and 1557 protein structures. However, enzymes from Basidiomycota in this database have not been characterized nor could a crystal structure be obtained [13]. Despite their strong impact on the functionality of lipases, not all lipases possess a lid. Furthermore, lids can be composed of one or more loops and/or α-helices [14]. By the use of single point mutations at the lid-domain, the chain length specificity of a lipase from *Pseudomonas fragi* could be increased towards octanoic acid, as shown by Santarossa et al. [15], which highlights the importance of the lid as a target for protein engineering (PE). Another promising approach is to modify the substrate channel to enhance the chain length specificity. Therefore, either the substrate channel can be blocked via steric hindrance by altering small amino acid residues to larger ones or by changing the protein conformation influencing the substrate channel. The steric hindrance approach was already successfully used as described by Schmitt et al. [16], where the chain length specificity of a lipase from the yeast *Candida rugosa* was modified by blocking the substrate channel with sterically demanding amino acids.

The aim of this study was to create PCI_Lip mutants with an optimized chain length specificity regarding the hydrolysis of triglycerides in order to improve its capability to replace PGE in the cheese-making process. For this purpose, PCI_Lip mutants were created in silico via a semi-rational design approach, expressed and analyzed in vivo, and in an initial micro-scale application test on Feta-type brine cheese. In addition, by investigating the effects of different mutations on the activity of the PCI_Lip, insights into suitable mutation strategies (e.g., lid vs. substrate channel), as well as the quality of the in silico prediction of relevant mutation positions for lipases from Basidiomycota can be gained.

## 2. Materials and Methods

### 2.1. Chemicals

Dodecanoic acid (99%) and tetradecanoic acid (99%) were obtained from Acros Organics (Geel, Belgium). Butyric acid (99%), hexanoic acid (99%), *p*-nitrophenyl octanoate (96%), and tetracycline were bought from Alfa Aesar (Karlsruhe, Germany). Coomassie^®^ Brilliant Blue R250 and tetramethylethylendiamine (TEMED) were purchased from AppliChem (Karlsruhe, Germany). Acetic acid (100%), agar-agar Kobe I, bovine albumin fraction V (98%), bromophenol blue (sodium salt), citric acid (99.5%), chloramphenicol (98.5%), decanoic acid (99%) disodium hydrogen phosphate (99.5%), ethanol (99.8%), glycine (99%), isopropanol (99.8%), kanamycin sulfate, lysogeny broth (LB) medium, *β*-mercaptoethanol, octanoic acid (99.5%) terrific broth (TB) medium, 2-[4-(2,4,4-trimethylpentan-2-yl)phenoxy]-ethanol (Triton X-100), tris(hydroxymethyl)-aminomethane (TRIS, 99%), TRIS-hydrochloride (99%), dipotassium hydrogen phosphate (99%), potassium dihydrogenphosphate (98%), ROTI^®^ mark standard, Rotiphorese^®^ Gel 40, sodium chloride (99.8%), sodium deoxycholate (98%), sodium dodecyl sulfate (99%), sodium dihydrogen phosphate (98%), and sodium hydroxide were provided by Carl Roth (Karlsruhe, Germany). Ammonium persulfate (100%) and hexadecenoic acid (100%) were bought from Merck (Darmstadt, Germany) and LB-SOC (Outgrowth medium) from New England BioLabs^®^ (Frankfurt am Main, Germany). Isopropyl-*β*-d-thiogalactopyranoside (IPTG) was purchased from Serva Electrophoresis (Heidelberg, Germany). Fastblue RR Salt, glycerol (99%), imidazole (99.5%), *p*-nitrophenyl acetate (98%), *p*-nitrophenyl butyrate (98%), *p*-nitrophenyl valerate (98%), *p*-nitrophenyl hexanoate (98%), *p*-nitrophenyl palmitate (98%), and l-(+)-arabinose (99%) were supplied by Sigma Aldrich (Schnelldorf, Germany). Gum arabic (X-E IRX 75200) was bought from Symrise (Holzminden, Germany) and hydrochloric acid (25%) from Th. Geyer (Renning, Germany). Helium (5.0) was purchased from Praxair (Düsseldorf, Germany) and nitrogen (5.0) from Air Liquide (Düsseldorf, Germany).

### 2.2. Microorganisms

*Escherichia coli* 10 beta cells were obtained from Zymo research Europe (Freiburg, Germany), and *E*. *coli* BL21 (DE3) cells were purchased from New England BioLabs^®^.

### 2.3. Construction of the Protein Models

Software Modeller v.9.25 (maintained by Ben Webb, University of California San Francisco, San Francisco, CA, USA) was employed to build 3D structures of the PCI_Lip wild type (WT) and the mutants by homology modeling [17]. The obtained structures were placed into a virtual water box (90 × 90 × 90 Å^3^ [1 Å = 10^−10^ m]) to solvate them. Afterwards, a molecular dynamic (MD) simulation was performed with the NAMD program v.2.14 (maintained by James C. Phillips, University of Illinois at Urbana-Champaign, Urbana-Champaign, IL, USA) in combination with the CHARMM36 force field [10].

### 2.4. Molecular Docking and Prediction of Relevant Positions

Molecular docking was carried out by AutoDock Vina software package v. 1.2.3 (maintained by Arthur J. Olson, The Scripps Research Institute, La Jolla, CA, USA). with the Broyden–Fletcher–Goldfarb–Shanno (BFGS) method [18]. The AutoDock tools package (maintained by Arthur J. Olson, The Scripps Research Institute, La Jolla, CA, USA) was employed to convert the PDB files of ligands and receptors into the relevant PDBQT files, which are the modified protein data bank format containing atomic charges, atom type definitions and, for ligands, topological information (rotatable bonds) [19]. The semi-flexible docking method was used. The threshold of the distance between the crucial carboxyl carbon atoms in six *p*-nitrophenol-(pNP)-esters with different chain lengths and the serine oxygen of S213 of the PCI_Lip active site was set as 6.5 Å. Other parameters were set as defaults. Based on the cavities in the active site, eight positions were predicted, seven in the channel and one in the lid, and selected as relevant positions. Unbiased, all eight relevant positions were swapped in silico with all 19 other amino acids. Then, molecular docking was carried out to screen this virtual mutant library (VML) by docking the aforementioned pNP-esters, while the same parameters were used as for the PCI_Lip wild type (WT).

### 2.5. In Silico Analysis of Diffusion and Binding of Triglycerides

The structures of triglyceride ligands (homo-triglycerides with three different carbon chains (C6:0, C10:0, C16:0)) were generated based on the standard CHARMM36 force field. All micro-units were built by Packmol program v. 20.1.0, which was maintained by Leandro Martínez (University of Campinas, São Paulo, Brazil), José Mario Martínez (University of Campinas, São Paulo, Brazil) and Ernesto G. Birgin (University of São Paulo, São Paulo, Brazil). The size of the water box was set as 90 × 90 × 90 Å^3^ [20]. The aqueous solution contained one molecule of PCI_Lip or L305A mutant and 16 substrate molecules. Next, an MD simulation (NAMD v.2.14 with standard CHARMM36 force field) was used to study their kinetic behaviors [21,22]. The temperature and pressure were set to 37 °C and 1 atm [23]. The MD simulation time was set as 240 ns.

### 2.6. Site-Directed Mutagenesis

After the transformation of chemically competent *E. coli* zymo 10 beta cells with a codon-optimized synthetic gene encoding PCI_Lip (BioCat, Heidelberg, Germany) in the pET-28a(+) vector as described by Green and Rogers [24], the plasmid was isolated from overnight cultures using the FastGene Plasmid Mini Kit (NIPPON Genetics, Düren, Germany). The primer design and PCR reaction setup for site-directed mutagenesis were based on the QuikChange™ protocol (Stratagene, La Jolla, CA, USA). A list of primers used in this study can be found in Appendix A. For PCR, 1 μL plasmid template, each 1 μL forward and reverse primer, 1 μL dNTPs, 10 μL 5× Phusion HF buffer and 0.5 μL Phusion DNA polymerase (New England BioLabs^®^, Ipswich, MA, USA) were mixed and adjusted to a total volume of 50 μL. To perform the PCR at different temperatures, aliquots of 10 µL volume were separated. The reaction was performed using a T100™ thermal cycler (Bio-Rad Laboratories, Feldkirchen, Germany) with the following program: initial denaturation for 1 min at 95 °C, denaturation for 45 s at 95 °C, annealing for 45 s at 60 to 73 °C, extension for 5 min at 68 °C (30 cycles), and final extension for 7 min at 68 °C. In order to digest the remaining template DNA, the PCR tubes were pooled, 1 µL DpnI (Thermo Fisher Scientific, Dreieich, Germany) was added, and the mixture was incubated for 2 h at 37 °C. The DpnI-digested PCR products were then directly used for the transformation of chemically competent *E. coli* 10 beta cells. The preparation of the competent cells and the transformation were performed as described by Green and Rogers [24]. The transformants were selected for kanamycin resistance on LB agar plates and respective colonies were picked and used for overnight cultures and plasmid isolation. For the confirmation of the mutations, the plasmids were sequenced by Eurofins Genomics (Ebersberg, Germany), and the results of the sequencing were analyzed with Geneious 9.1 (Biomatters, Auckland, New Zealand). Approved plasmids were used to transform *E. coli* BL21 (DE3) as described above. For soluble protein expression, *E. coli* BL21 (DE3) cells were transformed with the chaperon plasmid pG-KJE8 from TaKaRa BIO INC. (Kusatsu, Japan) for co-expression of the chaperones DnaK-DnaJ-GrpE and GroES-GroEL [25,26]. Cultures were maintained as cryostocks with 36% *v*/*v* glycerol at −80 °C.

### 2.7. Protein Expression and Purification

For the expression of the PCI_Lip WT and its mutants, 15 mL LB medium containing 25 µg mL^−1^ kanamycin and 20 µg mL^−1^ chloramphenicol were inoculated with 7.5 µL cryostock and incubated overnight at 37 °C and 180 rpm. For the main culture, 400 mL TB medium (with the same antibiotics and concentrations as in the overnight cultures) were inoculated with the whole overnight culture in a 2 L baffled shake flask. A total of 0.5 mg mL^−1^
l-(+)-arabinose and 5 ng mL^−1^ tetracycline were added to induce the expression of chaperones. The culture was incubated at 37 °C and 180 rpm to an OD_600_ of 0.8. Subsequently, the expression of the target protein was induced by adding 0.5 mM IPTG, and the temperature was reduced to 20 °C. After 20 h of incubation, the culture was harvested by centrifugation (3500× *g* at 4 °C, 10 min). The *E. coli* pellets were washed twice with 25 mL 80 mM potassium phosphate buffer, pH 7.0, and stored at −20 °C.

The *E. coli* pellets were resuspended in 5 mL 90% binding buffer (50 mM sodium phosphate buffer with 0.3 M sodium chloride at pH 7.5) with 10% elution buffer (binding buffer with the addition of 250 mM imidazol). After three times sonication for 2.5 min with a sonifier MS72 (microtip diameter 5 mm, cycle 5 and the amplitude 50%, Bandelin Electronic, Berlin, Germany,) on ice, the suspension was centrifuged (3500× *g* at 4 °C, 10 min). The supernatant was again centrifuged (14,000× *g* at 4 °C, 10 min) and subjected to immobilized metal ion affinity chromatography (IMAC). IMAC purification was performed at an NGC Quest 10 (Bio-Rad Laboratories) using a HisTrap^TM^ High Performance 5 mL Ni-NTA column (Cytiva Europe, Hoegaarden, Belgium). The IMAC method comprises the application of one column volume (CV) sample with a flow rate of 1.5 mL min^−1^ prior to two washing steps with the first 5% elution buffer for four CVs and then at 30% for another four CVs. The target protein was eluted with 100% elution buffer for two CVs.

After IMAC purification, the sample was desalted using three connected HiTrap^®^ Desalting columns (Cytiva Europe) with a CV of 5 mL each. Therefore, two CV desalting buffer (80 mM potassium phosphate buffer, pH 7.0) was used at a flowrate of 2 mL min^−1^. As the last purification step, a second size exclusion purification was performed in a centrifugal filter with a molecular mass cut-off of 30 kDa (Merck) by washing the samples up to ten times with desalting buffer. To control the successful expression of the PCI_Lip WT and the respective mutants, sodium dodecyl sulfate-polyacrylamide gel electrophoresis (SDS-PAGE) of the purified enzyme fractions was performed according to Laemmli [27] with a 4% stacking and 12% resolving gel. The gels were stained with Coomassie Brilliant Blue R 250. The purified sample was stored at 4 °C for further experiments.

### 2.8. Determination of Hydrolytic Activity

For the determination of the hydrolytic activity, photometric assays with different pNP-esters were performed [28,29]. pNP-acetate (pNPA), pNP-butyrate (pNPB), pNP-valerate (pNPV), pNP-hexanoate (pNPH), and pNP-octanoate (pNPO) were used as substrates for the detection of the esterase activity of the samples [28]. For this assay 130 µL potassium phosphate buffer (80 mM, pH 7.0) with 0.5% *v*/*v* Triton X-100 (pNPA without Triton X-100) were mixed with 20 µL enzyme solution in a 96-well micro-plate. To start the reaction, 50 µL of substrate solution with 3.5 mM of one of the five pNP esters was added. The absorbance at 405 nm was measured for 10 min at 30 °C using a Synergy 2 reader (BioTek, Friedrichshall, Germany). The lipase activity was measured similar to the esterase assay by using pNP-palmitate (pNPP) as the substrate. In this assay, 50 µL enzyme solution was mixed with 200 µL substrate solution (80 mM, pH 8.0, 2.2 g L^−1^ sodium deoxycholate, 1.1 g L^−1^ gum arabic) containing 0.4 mM pNPP. The reaction was performed at 37 °C for 15 min. One unit of enzymatic activity equals the amount of enzyme that produces 1 µmol of pNP per minute under assay conditions (esterase assay: *ε*_pNP_ = 0.00985 L µmol^−1^ cm^−1^ [28], lipase assay: *ε*_pNP_ = 0.0183 L µmol^−1^ cm^−1^ [29]).

### 2.9. Biochemical Characterization

For PCI_Lip WT and the mutants F91L, L302G and L305A, Michaelis-Menten kinetics and T_50_^60^ values were determined. For the Michaelis-Menten kinetics, different concentrations of pNPB, pNPH, pNPO and pNPP were used in the respective esterase/lipase assay. Every substrate concentration was measured in trpicates. The obtained data were analyzed via OriginPro 2021 by fitting the Y-data (category: enzyme kinetics, function: Michaelis-Menten, iteration algorithm: Levenberg–Marquardt, without weighting, equation: $y=\frac{v_{max}\;\cdot \;x}{K_{M}\;+\;x}$). The required protein concentration was determined by the Bradford assay Roti Nanoquant (Carl Roth) following the manufacturer’s manual.

The thermostability of the enzymes was examined by incubation of 100 µL enzyme solution at different temperatures for 60 min. Afterwards, the activity of the samples was measured in a photometric esterase assay using pNPO as the substrate. The measurements were performed in triplicates. The obtained Y-values were analyzed with a sigmoidal fit, whose turning point corresponds to the T_50_^60^ value, the temperature at which after treatment for 60 min, the activity of the enzyme is reduced by 50%. The graphical analysis was carried out with OriginPro 2021 (OriginLab, Northampton, MA, USA).

### 2.10. Application in Feta-Type Cheese Production and Sensory Evaluation

Application tests of the PCI_Lip WT and the mutants F91L, L302G and L305A were carried out to examine the effects on cheese flavor. For this purpose, 3 L full-cream milk (3.5% fat, pasteurized, homogenized; Stich Feinkäserei, Ruderatshofen, Germany), which will result in two cheese loaves of approximately 200 g each, were mixed with cream until a fat content of 4.2% was reached. Subsequently, starter cultures of lactic acid bacteria (0.03 UC of Lyofast M036L; Sacco SRL, Cadorago, Italy) and 1 U of PCI_Lip WT or the respective mutant (except 0.7 for L302G) were added to the milk, which had been preheated to 33 °C. The PCI_Lip dosage of 1 U (0.7 U) corresponds to its activity towards pNPO in the esterase assay. In addition, reference cheeses without the addition of lipases and with the commercial PGE opti-zym z10uc (0.35 g (10 L)^−1^; optiferm, Oy-Mittelberg, Germany) were produced. After pre-ripening for 45 min, calcium (0.3 mL of 34% *m*/*v* CaCl_2_, opti-calc; optiferm) and microbial rennet (0.75 mL opti-ren micro, 220 IMCU; optiferm) were added. The coagulated mass was cut into hazelnut-sized pieces (cubes with an edge length of approximately 15 mm) using a cheese harp. Afterwards, the curd was stirred manually, placed in prewarmed molds, and turned periodically to drain the whey. After a resting period of 24 h, the cheeses were put into brine (20% *m*/*v* table salt at pH 5.2, adjusted with 80% *m*/*v* lactic acid) for 50 min and drained. For conservation, the loaves were submerged into a natamycin bath (opti-cid, 2 g L^−1^; optiferm) for 1 min and drained again. For ripening, the loaves were shrink-wrapped and stored at 13 °C for 30 d. All cheeses were sensorily examined after 30 d of ripening. The sensory analysis was performed by five trained panelists from optiferm, who performed a simple descriptive test according to DIN 10964:2014-11 (German Institute for Standardization). In this test, simple attributes for the appearance, texture, smell, and taste were listed by the panelists. The cheese was presented in 1 × 1 cm cubes with single-digit codes. One loaf of the respective cheese mutant was used for the sensory evaluation; the second loaf was used for the analysis of vFFA by means of solid phase micro extraction gas chromatography-mass spectrometry (SPME-GC-MS) as described below.

### 2.11. SPME-GC-MS Analysis of vFFA

To examine the effects of different PCI_Lip mutants on the aroma of the cheeses, the vFFAs of the cheeses were analyzed by headspace SPME-GC-MS. Therefore, 3 g of cheese was transferred into a 20 mL headspace vial. After the addition of 2 mL HCl (2 M), the mixture was carefully homogenized via a vortex-mixer. The samples were incubated for 10 min at 55 °C (250 rpm agitation rate) by means of an MPS 2XL multipurpose sampler (GERSTEL, Mühlheim an der Ruhr, Germany) and subsequently extracted for 40 min (55 °C, 250 rpm) using a polydimethylsiloxane (PDMS)/divinylbenzene (DVB) SPME fiber (1 cm × 65 µm, Supelco, Steinheim, Germany). The analytes were desorbed in the GC inlet (SPME liner, 0.75 mm inner diameter, Supelco) at 250 °C for 90 s. After desorption, the SPME fiber was baked out for 5 min at 250 °C. For GC analysis, an Agilent (Waldbronn, Germany) 7890A gas chromatograph, equipped with a split/splitless (S/SL, split ratio 5:1) inlet and a polar VF-WAXms column (30 m × 0.25 mm, 0.25 µm film thickness, Agilent; temperature program: 40 °C (3 min), 5 °C min^−1^ to 240 °C (12 min); carrier gas: helium at 1.2 mL min^−1^ (constant)) was used. The GC system was connected to an Agilent 5975C mass spectrometer (transferline temperature: 250 °C, source temperature: 230 °C, quadrupole temperature: 150 °C, ionization energy: 70 eV, collision gas: nitrogen at 2.25 mL min^−1^, quencher gas: helium at 2.25 mL min^−1^, ms scan *m*/*z* 33–300). For substance identification, obtained mass spectra and calculated retention indices according to van den Dool and Kratz [30] were compared to those of authentic standards (C2:0, C4:0, C6:0, C8:0, C10:0, C12:0, C14:0, and C18:0 (40 mg L^−1^ in *n*-hexane) and the National Institute of Standards and Technology (NIST) MS Search (2011).

## 3. Results and Discussion

### 3.1. In Silico Analysis of the Relevant Positions

PCI_Lip belongs to the α/β-hydrolase fold enzyme family. Like all known α/β-hydrolases, it possesses the catalytically active core domain, a catalytic triad, which has been confirmed as S213, E339, and H449 by structural (Appendix A) and sequence alignment (Appendix A) [13,31,32]. The oxyanion hole signature is from GGGX-type, similar to that of *Candida rugosa* lipase 2 (PDB code of the closed structure 1GZ7) [32]. The substrate channel was predicted to be formed by the residues F91, F129, S163, L300, L302, A303, L305, I245, and I529, which were analyzed by investigating the cavities around the nucleophile S213 (Appendix A). These residues are different compared to *C. rugosa* lipase 2, which has a substrate-binding pocket formed by the residues L127, L132, and G450 (which corresponds to L131, M135, and A450 in PCI_Lip), and the entry of the hydrophobic channel is formed by the residues V296 and G344 (which correspond to I298 and H342 in PCI_Lip) [32]. In addition, the model of PCI_Lip has two regions (L86-F95 and L131-D138), which can be considered as covering lids on the entrance of the active pocket (marked in red in the Appendix A), while one lid-forming region is known for *C. rugosa* lipase 2. In 1GZ7, the lid domain is formed by a flap (residues P65–D94) that lies flat on the protein surface, and an important aromatic ring of F69 is buried in the hydrophobic pocket, while in PCI_Lip, this residue corresponds to L73, which is farther away from the predicted lid region in PCI_Lip. Interestingly, in PCI_Lip, F91 is potentially covering this role, as it is in the lid region, and the aromatic ring of this residue is located towards the substrate channel (Appendix A). The lid domain mutagenesis/exchange can not only modulate the interfacial activity, it may also influence the activity, substrate specificity, and thermostability of an enzyme [14]. This was shown for *C. rugosa* lipases 1 and 3 [33]. Since the substrate channel of PCI_Lip appears to be different from that of 1GZ7, a focus was on identifying residues that interact with potential substrates in the hydrolysis of milk fat triglycerides, which contain more than 400 different fatty acids [3]. As the modeling of such complex fat is very cumbersome, the selectivity of PCI_Lip and mutants thereof was reflected by their affinities for model substrates with different lengths of carbon chains. For this purpose, pNP-esters of different chain lengths were docked inside the homology model of PCI_Lip (Appendix A). Special attention was paid to the interaction of the substrates inside the binding pocket of PCI_Lip. The substrates pNPA, pNPB, pNPV, pNPH, and pNPO were able to fully enter the binding pocket of the WT enzyme, while the carbon chains of the substrates were bent, except pNPA, as the carbon chain is too short. A slight difference in the binding of the longer-chain chromogenic substrate pNPP was observed by in silico analysis, with the pNP-moiety located more at the solvent-accessible part of the substrate channel, possibly due to the higher steric hindrance of the carbon chain within the substrate channel. The binding analysis of the pNP-esters revealed that the residues inside the substrate channel could be divided into two regions: the left region containing the residues L300 and I529, and the right region consisting of F129, S163, I245, L302, A303, and L305. However, the A303 residue is the furthest away from the substrate channel cavity and orientated out of the channel and was therefore not mutated (Appendix A), while S163 is also farther away in the original model based on the crystal structure of 1GZ7 (Appendix A), but after relaxing in the virtual water box, the substrate channel is a bit more open, and S163 is pointing into that channel. In *C. rugosa* lipase 1, the substrate specificity was changed by directly blocking the substrate channel where the residues P246, L304, and L410 were identified to be the key residues. An exchange with bigger hydrophobic residues increased the steric hindrance for long-chain substrates. However, the corresponding residues in the PCI_Lip model are far away from the active site [16]. Unbiased, all eight positions (F91, F129, S163, I245, L300, L302, L305, and I529) were swapped in silico with all 19 other amino acids, and all six substrates were docked to the mutants (Appendix A), and their binding to a total of 152 mutants was analyzed to identify crucial residues for tailoring the size and shape of the binding pocket. For the mutants F91L, L302G, and L305A, the docking of pNP-esters with different chain lengths is exemplified in Appendix A. Considering the criteria that more suitable binding modes of small- to medium-chain length substrates should be found and the binding affinity of those variants should be increased for those substrates, compared to long-chain pNPP, 35 mutants were predicted to have a higher selectivity for short- to medium-chain fatty acids (Table 1). A similar approach was followed for the lipase from *Geobacillus thermoleovorans,* where the mutants were ranked and analyzed by their calculated binding affinity [34]. With the used workflow, the laboratory screening effort for the PCI_Lip mutants was thus reduced to about 23% compared to an approach where all eight positions were fully saturated. Through this smart but small library, a more effective analysis can be performed. A similar approach was used for increasing the enantioselectivity of a bacterial esterase towards tetrahydrofuran-3-yl acetate [35].

### 3.2. Expression of the Mutants and Characterisation of Their Hydrolysis Profiles

From the 35 predicted mutants, 24 mutants were successfully heterologously expressed and purified (Table 1 and Appendix A). Afterwards, their hydrolytic activities were investigated by different pNP-esters. The evaluation of the different PCI_Lip mutants revealed major differences in both their overall activities and their chain length specificities (Figure 1). The mutants at position F91 in the putative lid domain showed overall lower activities towards pNPP. Expressed as the ratio between the activity towards pNPO and pNPP, the mutants showed a higher selectivity of 1.3 for F91T and F91N, and up to 2.1 for the F91L and F91H mutants, in comparison to the ratio of pNPO/pNPP for the WT of 0.9. Apart from that, the activity against all substrates increased for the F91L and F91T mutants, whereas it decreased for the F91H and F91N mutants. Given its higher overall activity and the discrimination of pNPP, the F91L mutant showed the most promising results. The mutation of F129, which is placed in the substrate channel, led to a drastic loss of hydrolytic activity, resulting in almost inactive enzymes. Unfortunately, no mutant at position S163 could be solubly expressed, indicating that this amino acid may be crucial for the correct protein folding. Interestingly, the I245F mutant did not lead to an improvement in the pNPO/pNPP ratio but showed a significant increase in hydrolytic activity towards pNPB, pNPV, and pNPH and an overall increased hydrolytic activity compared to the WT. 

Mutations at the L300, L302, and L305 positions led to mutants with an improved hydrolysis profile, indicating that this part of the substrate channel is crucial for the chain length specificity of PCI_Lip. The L300R mutant showed an improved pNPO/pNPP ratio of 1.8 and an overall increase in activity, while the L300P and L300G mutants resulted in inactive or non-specifically hydrolyzing enzymes. The L302G and L302P mutants both revealed a decrease in activity against all tested substrates of about 50% compared to the PCI_Lip WT. These mutants had a major impact on the specificity of PCI_Lip. They showed almost no activity against pNPO and pNPP while having their highest activity against pNPH for the L302G and against pNPV for the L302P mutant. L305R had only a minor impact on the specificity as its hydrolysis profile mostly resembled that of the WT. The L305H and L305Y mutants showed a significant increase in hydrolytic activity against pNPP, leading to unfavorable pNPO/pNPP ratios of 0.6 for L305H and of 0.2 for L305Y (WT pNPO/pNPP: 0.9). Simultaneously, the overall activity was increased for the L305H mutant, while that of L305Y was decreased. More expedient changes in the hydrolysis profiles were observed for L305A, L305M, and L305N. All three mutants revealed an increased pNPO/pNPP ratio of 1.9 for L305A, 1.7 for L305M, and 1.6 for L305N. Similar to the L302 mutants, a decrease in the overall hydrolytic activities was observed in the esterase/lipase assays, with L305N showing the lowest activity. The I529A mutant showed little to no activity, while the I529D, I529G, and I529W mutants showed a comparable (I529G) or even increased (I529D/W) overall activity compared to the WT. The I529D and I529G mutants both showed a favorable increase in the pNPO/pNPP ratio of 2.4 for I529D and of 1.8 for I529G. Furthermore, these mutants showed increased selectivity for pNPH.

In conclusion, the mutations in the substrate channel of PCI_Lip led to mutants with the desired chain length specificity. Especially L302G had a major impact on the hydrolysis profile by shifting the maximal activity from pNPO towards pNPH. L305A showed a less drastic shift in its hydrolysis profile but drastically reduced activity towards pNPP. F91L, L302G, and L305A were selected for further biochemical characterization and application experiments.

### 3.3. Biochemical Characterization of Mutants

The selected mutants F91L, L302G, and L305A were further characterized by means of Michaelis-Menten kinetics for selected substrates (pNPB, pNPH, pNPO, pNPP) and by thermostability experiments (T_50_^60^). The obtained values (*v*_max_, *K_M_*, *k*_cat_, *k*_cat_/*K_M_*) for these mutants, as well as those of the WT, are listed in Table 2, also highlight differences in the hydrolytic activity of PCI_Lip WT and its mutants (Appendix A).

The F91L mutant showed increased *k*_cat_/*K_M_* and *k*_cat_ values for pNPB, pNPH, and pNPO, while providing a more favorable *k*_cat_/*K_M_* or even a decreased *k*_cat_ against pNPP, supporting the observed changes in the hydrolysis profile of this PCI_Lip mutant. The overall decreased hydrolytic activity of the L302G and L305A mutants is reflected in lower *k*_cat_ and *k*_cat_/*K_M_* values for these two mutants. The highest *k*_cat_ value of L302G was determined against pNPH, and a more favorable catalytic efficiency ratio of pNPP/pNPO (*k*_cat_/*K_M_* for pNPP divided through *k*_cat_/*K_M_* for pNPO) for the F91L and L305A mutants was observed in the kinetic data. Investigation of the thermostability showed no significant changes in the T_50_^60^ values of the different mutants compared to the value of (31.5 ± 0.7) °C of the PCI_Lip WT (Appendix A).

### 3.4. Analysis of Mutants and Molecular Dynamic Simulations

At least three positions, F91, L302, and L305, modulate the chain length specificity. Therefore, these sites were further analyzed. F91 has a double function; on the one hand, as a crucial residue inside the putative lid domain; on the other hand, it covers the entrance of the active pocket with its phenyl ring side chain. In general, phenylalanine in proteins has fewer rotamers compared to most other amino acids except for proline, tyrosine, and tryptophane and is therefore structurally more rigid compared to most other amino acids [36]. Moreover, in the PCI_Lip model, many loops were above F91, which further restricted the free space for structural changes of the F91 residue. In the calculated homology model in the equilibrium state, the phenyl ring is facing the substrate channel resulting in an enormous steric hindrance for binding substrates in the entry region of the channel. After mutation and virtual screening, F91L was confirmed as the best candidate (Appendix A), as further suitable binding options were identified for all chromogenic substrates, which is consistent with the determined activities towards pNP-esters of different chain lengths. For this variant, a shift of the highest activity from pNPP to pNPO was observed in the activity profile. The aromatic ring structure has been replaced by a short-chain branched amino acid that is still able to facilitate hydrophobic interactions with the carbon chains of the substrates while creating more space. This allowed the substrates to bind deeper in the binding pocket (Appendix A). In Lip12 from *Yarrowia lipolytica*, which has no structural similarity with PCI_Lip, an exchange of F148L in the substrate channel could increase the catalytic efficiency for all tested substrates [37].

Within the substrate channel, the replacement of L302 and L305 had the greatest impact on the hydrolysis profiles. Based on the docking analysis, L302G was confirmed as the best candidate at this position, where the overall profile of the binding pocket was not significantly altered as shown in Appendix A, but the spatial arrangement of L305 was slightly altered, perhaps due to the reduced possibilities of hydrophobic interactions between the isobutyl groups of L302 and L305. Thus, the replacement of L by G directly reduces the relevant steric hindrances in the binding region. However, the activity against pNPP as the long-chain substrate was decreased, which might be an effect of less hydrophobic interactions in the L302G mutant compared to the WT enzyme. Similar effects are observed for the different isoforms of *C. rugosa* lipases, where weaker interactions with the substrates in lipase 2 occur compared to lipase 1 or lipase 3. Domínguez de María et al. [38] speculated that this phenomenon confers higher esterase behavior for lipase 2 compared to lipases 1 and 3, respectively.

In contrast, the exchange of L305A caused an obvious change in the binding pocket of the homology model of PCI_Lip. Due to the exchange, which results in a smaller space requirement for the residue L305, no interactions between F129 and L300 could be formed (Appendix A). Therefore, the space on the right part of the binding pocket, which was the main binding region in the WT enzyme, was reduced. In addition, F91 was removed from the binding pocket and turned towards the outer surface. These results were all beneficial in creating enough space in the left side of the binding pocket of the L305A mutant. The end of the lid, the residues L86–F95, was also affected by this mutation. L86–T89 were merged into a part of an α-helix, which might tighten the lid structure, and twisted it out to the reaction medium, which at the same time enlarged the entrance region of this active pocket. However, after docking, no pNPP molecule was observed in the right position. As the hydrolysis of long-chain fatty acids from triglycerides caused off-flavors in the final cheese product, this mutant is highly interesting.

To better understand the effects of the mutation, molecular dynamics simulations were performed with “model” homo triglycerides with fatty acids of chain lengths C6:0, C10:0, and C16:0, and the L305A mutant was compared to WT PCI_Lip. For this purpose, the changes of RMSD (Root Mean Squared Error) of the crucial distances between the carboxyl carbon atom (from different fatty acid chains in triglycerides) and the hydroxyl group oxygen of S213 in WT PCI_Lip or L305A mutant were analyzed (Appendix A).

Generally, the L305A mutant showed a higher binding affinity for the triglycerides than PCI_Lip WT, which was reflected by the number of suitable binding modes and residence time of fatty acid chains in the active center (Appendix A). For the wild type, most fatty acids entering the active pocket were long-chain fatty acids (C16:0). After mutation, the number of medium-chain fatty acids (C6:0 and C10:0) entering the active pocket was increased markedly. The ratio of these three fatty acids was also improved, indicating a higher selectivity of L305A. This effect might be due to the reshaping of the active pocket generated by the L305A exchange. However, not only the selectivity was changed, based on the MD analysis, but also the binding time of the substrates was different (Appendix A). Here, L305A was able to accommodate more fatty acid chains at the same time compared to PCI_Lip WT, possibly due to the longer time required for the interfacial activation of PCI_Lip WT, where the ligands were bound only in the later phase of the calculation.

### 3.5. Application of the Mutants in Cheese Production

All three optimized mutants were used in a micro-scale approach to Feta-type brine cheese production, followed by sensory and SPME-GC-MS analysis. The cheeses were produced with the addition of 1 U PCI_Lip WT, F91L, L305A, and 0.7 U of L302G. The sensory evaluation after 30 days of ripening showed differences primarily regarding the taste category rather than in the texture or the smell (Table 3). While a reference without the addition of any lipases showed neutral sensory properties, the reference produced with opti-zym z10uc possessed the desired piquant, goaty smell and taste.

The three cheeses produced with PCI_Lip mutants had different sensory properties among themselves and in comparison to the WT. While the WT was described as comparable to the opti-zym z10uc reference, the L305A mutant had a more intense taste and possessed a more pleasant acid profile than the cheeses produced with PCI_Lip or other mutants. Overall, the cheese with the L305A mutant was ranked as the best cheese in this initial trial. The SPME-GC-MS analysis revealed vFFA profiles that support the results of the sensory evaluation. Even though the differences between the WT and the mutants are smaller than indicated by the photometric assays, especially for F91L, shifts towards medium-chain vFFAs could be observed for L302G and L305A. The cheese produced with L305A showed a vFFA profile that resembles that of the reference cheese obtained with opti-zym z10uc regarding the ratios between the different vFFAs. These findings are consistent with the L305A mutant cheese being ranked as the most appealing during this study (Figure 2).

## 4. Conclusions

The first approach of an engineered lipase from Basidiomycota to modulate the chain length specificity was demonstrated. Through a small but smart library, the effort of expression, purification, and characterization was reduced by 77% (35 of 152 all possible mutants), while 18 mutants (roughly 50% based on the predicted) displayed a change in the hydrolysis profile, and 12 of these mutants (34%) had the desired effect of reduced activity against long-chain substrates. Three mutants that showed a drastic decrease in the activity towards long-chain substrates were subjected to cheese production and further analyzed. In initial application tests, the L305A mutant showed promising results to produce vFFA in brine cheese, based on the GC-MS measurements, similar to an animal PGE alternative. The cheese produced with L305A mutants was ranked the most appealing cheese in the sensory evaluation. Summarizing the results of this study, promising mutants of the PCI_Lip could be created. Especially the L305A mutant has a great potential to enable the production of cheeses, which have traditionally been produced with the aid of animal-derived lipases, under vegetarian, halal, or kosher conditions and, at the same time, to maintain their characteristic profile. Upcoming industrial pilot scale tests will have to validate the industrial usability of the novel enzyme.

## Figures and Tables

**Figure 1 foods-11-02608-f001:**
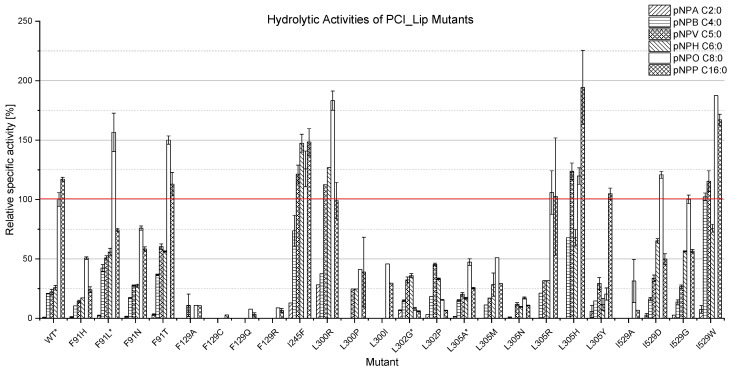
Relative specific activity [%] of PCI_Lip mutants (expression: *n* = 1, mutants marked with * *n* = 3) against selected model substrates. The relative specific activities were based on the specific activity [U mg^−1^] of the PCI_Lip WT against pNPO (set to 100%, red line). The photometric esterase/lipase assays were performed in triplicates. The error bars represent the standard deviation of these measurements.

**Figure 2 foods-11-02608-f002:**
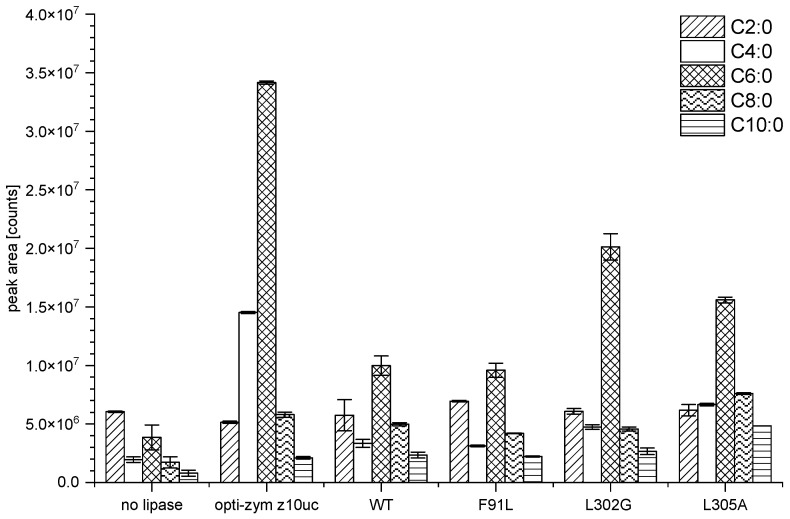
Feta-type brine cheese produced without addition of lipase, the reference PGE opti-zym z10uc, PCI_Lip WT (1 U), F91L (1 U), L302G (0.7 U), and L305A (1 U) after 30 d of ripening at 13 °C. Shown are peak areas of selected vFFA from the cheese samples after SPME-GC-MS analysis. The measurements were performed in duplicates and the error bar corresponds to the mean deviation. C2:0 = acetic acid, C4:0 = butyric acid, C6:0 = hexanoic acid, C8:0 = octanoic acid, and C10:0 = decanoic acid.

**Table 1 foods-11-02608-t001:** PCI_Lip mutants with higher selectivity towards short- and medium-chain fatty acids predicted in silico. x = successful expression, - = expression was not possible as no protein band was visible in the SDS-PAGE after purification.

Amino Acid	Mutant	Expression	Amino Acid	Mutant	Expression
F91	G	-	L300	I	x
	H	x		P	x
	L	x		R	x
	N	x		Q	-
	T	x	L302	F	-
F129	A	x		G	x
	C	x		P	x
	M	-	L305	A	x
	Q	x		H	x
	R	x		M	x
S163	H	-		N	x
	M	-		R	x
	P	-		Y	x
	V	-	I529	A	x
	Y	-		D	x
I245	F	x		E	-
	W	-		G	x
				W	x

**Table 2 foods-11-02608-t002:** Overview of the kinetic data of PCI_Lip WT and mutants F91L, L302G, and L305A against selected model substrates. Listed are the maximum velocity (*v*_max_), the Michaelis-Menten constant (*K_M_*), the turnover number (*k*_cat_), the catalytic efficiency (*k*_cat_/*K_M_*), and the protein concentration in all kinetic experiments, which were used for the calculation of *k*_cat_.

	Protein Concentration [µmol L^−1^]	Substrate	*v*_max_ [µmol min^−1^ L^−1^]	*K_M_* [mmol L^−1^]	*k*_cat_ [s^−1^]	*k*_cat_/*K_M_* [s^−1^ mol^−1^ L]
WT	1.96	pNPB	50.12	0.647	0.43	658
		pNPH	78.89	0.549	0.67	1222
		pNPO	215.82	0.718	1.83	2555
		pNPP	103.52	0.022	0.88	40,177
F91L	2.74	pNPB	115.89	0.786	0.71	898
		pNPH	119.80	0.504	0.73	1448
		pNPO	655.00	1.328	3.99	3005
		pNPP	136.65	0.042	0.83	20,024
L302G	3.82	pNPB	77.53	0.756	0.34	447
		pNPH	223.05	1.214	0.97	801
		pNPO	42.82	0.706	0.19	264
		pNPP	13.87	0.013	0.06	4733
L305A	4.60	pNPB	77.12	2.299	0.28	122
		pNPH	45.51	0.645	0.16	256
		pNPO	97.42	0.735	0.35	481
		pNPP	33.27	0.022	0.12	5598

**Table 3 foods-11-02608-t003:** Results of the sensory evaluation of different Feta-type brine cheeses produced with the addition of the reference PGE opti-zym z10uc, PCI_Lip WT (1 U), F91L (1 U), L302G (0.7 U), and L305A (1 U) and of a cheese without the addition of lipase after 30 days of ripening. The sensory properties were analyzed by five trained panelists at optiferm.

	No Lipase	Opti-Zym z10uc	WT	F91L	L302G	L305A
Appearance	Typically cream-colored, crumbly					
Texture	Firm, dry	Firm, dry	Softer/creamier than reference	Softer/creamier than reference	Firm, dry	Firm, dry
Smell	Neutral, milky	Typical for goat lipase, intense, aromatic, goaty, strong, comparable to parmesan	Comparable to opti-zym	Comparable to opti-zym; slightly off	Comparable to opti-zym; slightly off	Comparable to opti-zym; most pleasant and intense smell within this trial
Taste	Neutral, slightly sour	Typical for goat lipase, intense, aromatic, piquant, goaty, persistent aftertaste	Comparable to opti-zym	At first comparable to opti-zym but less intense; slightly off, acid taste differing from WT	At first comparable to opti-zym but less intense; slightly off, acid taste differing from WT, slightly rancid/bitter	Comparable to opti-zym, more intense then wildtype, pleasant acid profile, best taste within this trial

## Data Availability

All data generated or analyzed during this study are either included in this published article or can be found in the Appendix A.

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
