# Peer review of "Altering the Chain Length Specificity of a Lipase from Pleurotus citrinopileatus for the Application in Cheese Making"

_foods, 2022, doi:10.3390/foods11172608_

Round 1

Reviewer 1 Report

In Line 136, 158: The given value (angstrom) is not an International System (SI) or metric unit. According to the requirements of the Journal: SI Units (International System of Units) should be used. Imperial, US customary and other units should be converted to SI units whenever possible. So please write unit down correctly throughout the manuscript and supplementary.

In Line 263: Correct statement - "cheese loaves" to be more precise. The term loaves refers to bread. Please refer to the entire manuscript.

In Line 272: Correct the sentence "The coagulated mass was cut into hazelnut-sized pieces using a cheese harp". Provide specific dimensions. What does hazelnut-sized pieces mean? The hazelnut is spherical and can be of various sizes.

A little advice Authors may apply in this manuscript. It is recommended to emphasize the results for the dairy industry and/or consumer in a separate paragraph (not necessary, but it is worth highlighting it for greater *Interest to the readers).

Author Response

Reviewer 1

In Line 136, 158: The given value (angstrom) is not an International System (SI) or metric unit. According to the requirements of the Journal: SI Units (International System of Units) should be used. Imperial, US customary and other units should be converted to SI units whenever possible. So please write unit down correctly throughout the manuscript and supplementary.

Thank you for pointing out that we did not explain the “unit” angstrom. Of course, angstrom is not an SI unit, but it is still used in the field of structure determination, molecular modeling and enzyme engineering. As this part is especially of interest for researches active in this fields, we would prefer keeping “angstrom”, but we have added an explanation in line 146 (1 Å = 10-10 m).

In Line 263: Correct statement - "cheese loaves" to be more precise. The term loaves refers to bread. Please refer to the entire manuscript.

In fact, the term “loaf” is well established in the dairy field and is not limited to bread. An alternative term, namely “cheese wheel” does not perfectly match in our case because of the rather small-sized cheeses that were produced here.

Even in EU commission regulations for entering a designation in the register of protected designations of origin and protected geographical indications, the term “loaf” is used, e.g. in the requirements for Gouda:

(https://eur-lex.europa.eu/LexUriServ/LexUriServ.do?uri=OJ:L:2010:317:0022:0029:EN:PDF) [Herein: Point 4.2 Description – characteristic properties – line 1.]

The term “loaf” is also listed in the “dairy vocabulary” (VV-GmbH Volkswirtschaflicher Verlag München, revised edition of Casalis/Mann/Schulz, 1988).

For the given reason, the authors tend to stick to the term “loaf”.

In Line 272: Correct the sentence "The coagulated mass was cut into hazelnut-sized pieces using a cheese harp". Provide specific dimensions. What does hazelnut-sized pieces mean? The hazelnut is spherical and can be of various sizes.

In cheese-making, the term “hazelnut-sized” is well established to describe the approximate size of the curd particles. The cutting of the curd using a cheese harp is done by hand. Thus, terms like “hazelnut-sized” or “walnut-sized” are only rough guidelines for the cheesemaker.

The term “hazelnut” is also listed in the “dairy vocabulary” (VV-GmbH Volkswirtschaflicher Verlag München, revised edition of Casalis/Mann/Schulz, 1988).

For the given reason, the authors tend to stick to the term “hazelnut-sized”, but the approximate metric dimensions “(cubes with an edge length of approximately 15 mm)” (in lines 282-283) have been added to addresse the reviewer’s concerns.

A little advice Authors may apply in this manuscript. It is recommended to emphasize the results for the dairy industry and/or consumer in a separate paragraph (not necessary, but it is worth highlighting it for greater *Interest to the readers).

The authors are very grateful for this advice.

The results for the dairy industry and the consumer of traditional cheese specialties are included in the conclusions (in line 578-583) of the manuscript now:

“Summarizing the results of this study, promising mutants of the PCI_Lip could be created. Especially the L305A mutant has a great potential to enable the production of cheeses, which have traditionally been produced with the aid of animal-derived lipases, under vegetarian, halal or kosher conditions and, at the same time, to maintain their characteristic profile. Upcoming industrial pilot scale tests will have to validate industrial usability of the novel enzyme.”

Reviewer 2 Report

The topic of the article falls within the thematic scope of the journal FOODS.

Very interesting and inspiring research. The search for alternative solutions for the dairy industry, thanks to which it will be possible to offer products to consumers with special requirements (vegetarians, kosher, etc.) is very important at present. It also seems that the developed lipolytic preparation can also be used to obtain new types of cheeses with slightly different sensory features and possibly to accelerate the ripening of the cheeses.

I have only one comment on the manuscript. Namely, at the end of the Introduction chapter, there is no research objective clearly stated and formulated in a separate paragraph.

The manuscript is prepared very carefully and does not require any changes apart from the above mentioned and adding bibliographic data for one of the cited publications (line 653).

All comments and suggestions for corrections were introduced in the review mode to the attached pdf file.

Author Response

Reviewer 2

The topic of the article falls within the thematic scope of the journal FOODS.

Very interesting and inspiring research. The search for alternative solutions for the dairy industry, thanks to which it will be possible to offer products to consumers with special requirements (vegetarians, kosher, etc.) is very important at present. It also seems that the developed lipolytic preparation can also be used to obtain new types of cheeses with slightly different sensory features and possibly to accelerate the ripening of the cheeses.

I have only one comment on the manuscript. Namely, at the end of the Introduction chapter, there is no research objective clearly stated and formulated in a separate paragraph.

As suggested by the reviewer, a paragraph summarizing the aims of our study has been added at the end of the “Introduction” chapter. We added in the lines 97-105:

“The aim of this study was to create PCI_Lip mutants with an optimized chain length specificity regarding the hydrolysis of triglycerides in order to improve its capability to replace PGE in the cheese making process. For this purpose, PCI_Lip mutants were created in silico via a semi-rational design approach, expressed and analyzed in vivo and in an initial micro-scale application test on Feta-type brine cheese. In addition, by investigating the effects of different mutations on the activity the PCI_Lip, insights in suitable mutation strategies (e.g. lid versus substrate channel) as well as on the quality of the in silico prediction of relevant mutation positions for lipases from Basidiomycetes can be gained.”

The manuscript is prepared very carefully and does not require any changes apart from the above mentioned and adding bibliographic data for one of the cited publications (line 653).

The bibliographic data for the cited publication in line 665-666 have been added.

All comments and suggestions for corrections were introduced in the review mode to the attached pdf file.

Reviewer 3 Report

This paper researched the altering the chain length specificity of a lipase from Pleurotus  citrinopileatus for the application in cheese making. The research is interesting and usefule, however, paper still need revised. 

There is no clear aim of the work.

L99-102 In the introduction, the Authors write about the advantages of the L305A mutant of PCI_Lip before conducting the research?

The results obtained regarding lipase have been described in great detail, but it is difficult to pick out the most relevant information. It is a pity that the Authors did not show such involvement when verifying the impact of the identified mutants on the quality of the cheese. It is a serious failure to conclude on the basis of the analysis of a single sample. Only 1 sample of each type of cheese was analysed. This is far not enough to conclude on the possibility of selecting the best option of the enzyme mutant of PCI_Lip.

Where do the Authors see similarities in the vFFA of the L305A mutant cheese and the reference PGE optizyme 547 z10uc? (L542)

Author Response

Reviewer 3

This paper researched the altering the chain length specificity of a lipase from Pleurotus  citrinopileatus for the application in cheese making. The research is interesting and usefule, however, paper still need revised. 

There is no clear aim of the work.

Thank you for that comment, we added a paragraph summarizing the aims of our study at the end of the “Introduction” chapter (cf. response to reviewer 2).

L99-102 In the introduction, the Authors write about the advantages of the L305A mutant of PCI_Lip before conducting the research?

L99-102 was meant to be a slight preview into to results of this study to enhance the readers interest, nonetheless this part was deleted and a separate paragraph describing the aims of our work was added at the end of the “Introduction” chapter.

We wrote in lines 97-105:

“The aim of this study was to create PCI_Lip mutants with an optimized chain length specificity regarding the hydrolysis of triglycerides in order to improve its capability to replace PGE in the cheese making process. For this purpose, PCI_Lip mutants were created in silico via a semi-rational design approach, expressed and analyzed in vivo and in an initial micro-scale application test on Feta-type brine cheese. In addition, by investigating the effects of different mutations on the activity the PCI_Lip, insights in suitable mutation strategies (e.g. lid versus substrate channel) as well as on the quality of the in silico prediction of relevant mutation positions for lipases from Basidiomycota can be gained.”

The results obtained regarding lipase have been described in great detail, but it is difficult to pick out the most relevant information. It is a pity that the Authors did not show such involvement when verifying the impact of the identified mutants on the quality of the cheese.

The results of the “Protein Engineering” part include a large amount of data, which cannot be summarized easily. Nevertheless, we have tried to clearly summarize the main focus of our study as indicated above in our response to reviewers 1 and 2.

It is a serious failure to conclude on the basis of the analysis of a single sample. Only 1 sample of each type of cheese was analysed. This is far not enough to conclude on the possibility of selecting the best option of the enzyme mutant of PCI_Lip.

First of all, we fully agree with the reviewer that no final statement on the industrial applicability of these PCI_Lip mutants can be based on the analysis of a single cheese batch. However, all of the cheese samples were analyzed in duplicates, and the main focus of this study was the protein engineering of this basidiomycetous lipase. The application test was performed to obtain first information about the principal practicability of the PCI_Lip mutants. In our future research, the industrial applicability point will be of central interest. To tone done the overall claims of the manuscript, the formulations on the application test were changed to clarify the preliminary character of the cheese making process throughout the manuscript.  E.g.,  “in Feta-type brine cheese production, …” was changed to “in a micro-scale approach of Feta-type brine cheese production, …” or

The conclusion was changed in lines 574-576 from

“The L305A mutants was pointed out to be the best candidate to produce free fatty acids in brine cheese based on the GC-MS measurements, similar to an animal pregastric esterase alternative.”

to

“In initial application tests, the L305A mutant showed promising results to produce free fatty acids in brine cheese, based on the GC-MS measurements, similar to an animal pregastric esterase alternative.”

In line 578-583:

“Summarizing the results within this study promising mutants of the PCI_Lip could be created and the L305A mutant has a great potential to enable the production of cheeses, which are traditionally produced with the aid of animal lipases under vegetarian, halal or kosher conditions and, at the same time, to maintain their characteristic profile. While in upcoming industrial pilot microscale work the usability has to be validated.”

Where do the Authors see similarities in the vFFA of the L305A mutant cheese and the reference PGE optizyme 547 z10uc? (L542)

Similarities can be seen considering the overall distribution of vFFA with a maximum for C6:0 decreasing constantly towards shorter and longer chained vFFA. Of course, the vFFA concentrations between the L305A mutant cheese and the reference are not similar but the overall profile resembles that of the PGE reference.

Round 2

Reviewer 3 Report

Dear Authors,

in my opinion the manuscript has been sufficiently improved and can be published in Foods.